# Speaker Verification Employing Combinations of Self-Attention Mechanisms

Ara Bae and Wooil Kim * 

Department of Computer Science and Engineering, Incheon National University, Incheon 22012, Korea;
arbae@inu.ac.kr
* Correspondence: wikim@inu.ac.kr; Tel.: +82-32-835-8459

**Abstract:** One of the most recent speaker recognition methods that demonstrates outstanding performance in noisy environments involves extracting the speaker embedding using attention mechanism instead of average or statistics pooling. In the attention method, the speaker recognition performance is improved by employing multiple heads rather than a single head. In this paper, we propose advanced methods to extract a new embedding by compensating for the disadvantages of the single-head and multi-head attention methods. The combination method comprising single-head and split-based multi-head attentions shows a 5.39% Equal Error Rate (EER). When the single-head and projection-based multi-head attention methods are combined, the speaker recognition performance improves by 4.45%, which is the best performance in this work. Our experimental results demonstrate that the attention mechanism reflects the speaker's properties more effectively than average or statistics pooling, and the speaker verification system could be further improved by employing combinations of different attention techniques.

**Keywords:** speaker verification; self-attention; attention combinations

---

## 1. Introduction

The key to good speaker recognition systems lies in generating speaker features that can effectively distinguish different speakers. Conventional speaker recognition systems used spectral representations such as linear predictive coefficients (LPC) or Mel-frequency cepstral coefficients (MFCC) for speaker feature and Gaussian Mixture Model (GMM) for speaker modeling [1–3]. The i-vector feature, which was introduced in the early 2010s, is one of the most robust speaker features used until recently [2,4]. Studies using deep neural networks have been actively conducted since the mid-2010s. These studies mainly used speaker features extracted from recurrent neural network (RNN) series models such as the time-delay neural network and long short-term memory (LSTM). Acoustic feature such as MFCC or Mel-filter bank outputs are used as input to the deep neural model and a fully connected layers are usually added to the model. Average or statistical pooling is applied to the model or the output stage to convert frame-level representation into utterance-level representation, and the embedding is obtained as speaker features [5–9].

From the perspective of a speaker recognition system, there may be some parts-of-speech signals that have more critical information for feature creation than the other parts. In addition, a simple average over the entire duration may degrade the original speaker characteristics. To solve this problem, the attention mechanism [10] was introduced in the speaker recognition system to generate features that focus more on the critical information [11,12]. As a result, excellent performance could be achieved by letting the features focus on the input word having the closest correlation with the translated word in machine translation. The speaker recognition performance was further improved by employing the attention layer rather than the simple average technique, proving that the attention

method is more effective at focusing on the speech frames needed for generating the features that effectively express the speaker's characteristics.

Multi-head attention, which performs attention mechanism in parallel in the subspaces of features, is known to be more effective than traditional methods [13–19]. Previous studies have reported that the multi-head attention model judges the importance of multiple people from various perspectives, while in the single-head attention model, the importance is judged based on a single person's viewpoint.

In this paper, we investigate the effectiveness of the basic LSTM-based embedding extractor model without an attention layer and a single/multi-head attention layer. The multi-head attention methods might lose essential speaker features of the frame-level through the process of projection or splitting. The same issue was raised by [20] that the possible correlations among the sub-vectors created by the multi-head attention were ignored when calculating the attention weights. To solve this issue and improve the speaker recognition performance by leveraging the attention methods, we propose two types of combination methods for generating the new embedding: (1) a combination of different types of multi-head attentions and (2) a combination of single-head and multi-head attention methods.

Section 2 describes the definition of the embedding extractor and attention layer based on LSTM. The two types of combination methods proposed are presented in Section 3. The experimental setting and results are discussed in Section 4, and our work is concluded in Section 5.

## 2. Related Works

The i-vector, which is a feature based on the Gaussian mixture model (GMM)–universal background model (UBM) model, has demonstrated robust speaker recognition performances in many studies. Therefore, we compared the i-vector with a speaker embedding using the proposed methods based on a neural network. As deep neural networks are extensively applied for speech processing tasks, various features such as the d- and x-vectors have also been introduced. We herein focus on the speaker embedding extracted using the LSTM model. As the vanilla RNN model has a critical problem wherein the past information is lost over time (i.e., vanishing gradient), the LSTM model demonstrates better performance for the temporal sequence data. To train the embedding extraction model, classification methods with the speaker ID as output and the Generalized End-To-End (GE2E) method for calculating a similarity matrix between speaker features are widely used. Unlike the conventional loss functions which calculate an error of the speaker ID classifier using speaker embedding, the GE2E constructs a mini batch with similarity score matrix by selecting multiple speakers and utterance. In this study, the model was trained using the GE2E loss [21].

### 2.1. LSTM-Based Embedding Extractor

The acoustic features extracted from the speech signal were used as input to the neural network. The frame-level features were aggregated to obtain speaker characteristics at the utterance-level. In the LSTM, the last frame contains a considerable amount of information because the previous state affects the next state. Therefore, the last frames of the output layer $h_t$ were averaged, and the averaged result was termed as an embedding. The framework with an embedding vector $e$ generated from the input acoustic feature $x$ is presented in Figure 1. $T'$ denotes the number of frames entering the LSTM as a result of the acoustic feature $x$ with the shifting windows, and $d$ is the output dimension.

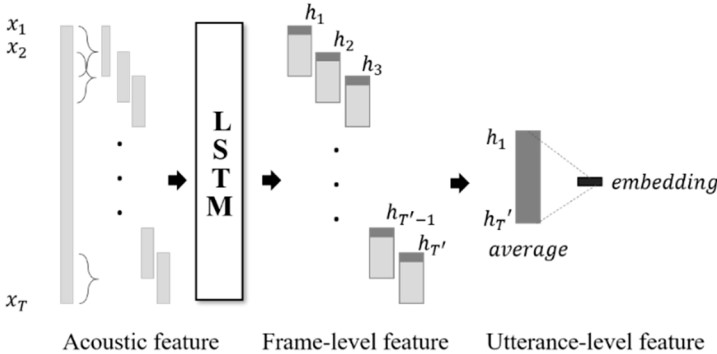

**Figure 1.** Extractor of LSTM-based speaker embedding framework.

## 2.2. Single-Head Attention

The single-head attention method obtains the weight value using shared parameters through a nonlinear attention layer, demonstrating better speaker recognition performance than the basic attention layer. The basic attention method calculates the weight by applying the output of the LSTM directly into the nonlinear activation function [11]. For the nonlinear activation function in Multi-Layer Perceptron (MLP) layer, a tanh function was used. $W \in \mathbb{R}^{d \times d}$ and $b$, $u \in \mathbb{R}^d$ were the trained parameters. As seen in Figure 2, the LSTM output $h$ was calculated with $W$ and $b$ and then placed into the tanh function, and finally multiplied with $u$ to generate $v$. The attention weight $\alpha_t$ was calculated using the softmax function with $v_t$, indicating the importance of the frames for representing speaker characteristics. The embedding vector $e$ was obtained weighted average by $h_t$ and $\alpha_t$, where the scalar $\alpha_t$ was broadcast by $d$-dimension.

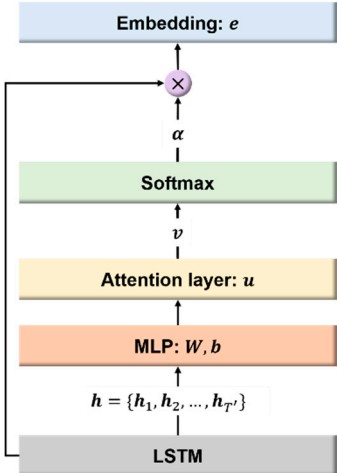

**Figure 2.** Block diagram of speaker embedding extraction using single-head attention; $\alpha$, $v$, and $h$ denote attention weight, attention layer output, and LSTM output respectively.

## 2.3. Multi-Head Attention

Multi-head attention methods produce as many attention weights as the number of heads $H$. Two types of the multi-head attention methods have been developed: one to project the frame-level features into the subspaces as the number of heads [15,17] and the other to split the frame-level features [16] as presented in Figure 3. The following sections present the calculation of the attention weight in each type of multi-head attention method. The principal difference between the two multi-head attention methods is that the split method creates sub-vectors by dividing the LSTM output as inputs to the sub-MLP layers and the projection method generates sub-vector as outputs of the MLP layer.

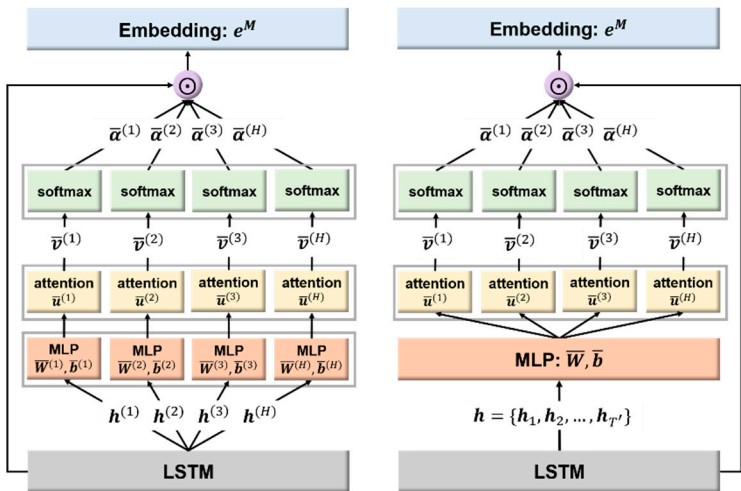

**Figure 3.** Block diagrams of speaker embedding extraction using multi-head attention with split (**left**) and multi-head attention with projection (**right**); $\overline{\alpha}^{(i)}$, $\overline{v}^{(i)}$, and $h$ denote attention weight, attention layer output, and LSTM output respectively.

The weights $\overline{\alpha}_t^{(i)}$ in Figure 3 for the frame $t$ of the $i$th head were obtained as $d/H$ dimensions and concatenated into a single vector. Here, $i = [1, 2, \ldots, H]$ indicates the index of the head. Subsequently, a $d$-dimensional vector $\alpha_t^M$ was generated; this vector shows the importance weight of multi-head attention methods as presented in Equation (1), where M represents "Multi-head".

$$
\alpha_t^M = \begin{bmatrix} \alpha_{t,1,\ldots,\frac{d}{H}}^M = \overline{\alpha}_t^{(1)} \\ \alpha_{t,\frac{d}{H}+1,\frac{d}{H}+2,\ldots,2\frac{d}{H}}^M = \overline{\alpha}_t^{(2)} \\ \vdots \\ \alpha_{t,(H-1)\frac{d}{H}+1,(H-1)\frac{d}{H}+2,\ldots,d}^M = \overline{\alpha}_t^{(H)} \end{bmatrix}
\tag{1}
$$

For example, when $d = 256$ and $H = 4$, each head's scalar weight value was copied to each element of a 64-dimensional vector and $\alpha_t^M$ became a 256-dimensional vector. In contrast to the single-head attention, the importance weight $\alpha_t^M$ of the multi-head attention method was in a vector; therefore, the embedding vector $e^M$ was obtained as follows:

$$
e^M = \sum_{t=1}^{T'} \alpha_t^M \odot h_t
\tag{2}
$$

Here, the operator $\odot$ denotes an element-wise multiplier for the two vectors $\alpha_t^M$ and $h_t$. For the single-attention method, the embedding was obtained by applying the same weight value to all elements of each LSTM output $h_t$. However, as presented in this section, the multi-head attention method generated the embedding by applying different weight values from the number of heads to the different parts of the LSTM output $h_t$.

## 3. Proposed Methods

We believe that the performance of the embedder in a speaker recognition system can be improved by leveraging the different aspects of single-head attention and the two different multi-head attention methods. To this end, we propose the use of two techniques.

### 3.1. Multi-Head Projection and Split Combination

In multi-head attention methods, multiple subspaces are created by projecting or splitting frame-level features. The attention weight is calculated from the trainable parameters $W$, $b$ and $u$ in

the same way as single-head attention. The significant difference from the single-head attention is different number of parameters of each head used for calculation, so generates attention weights as many as the number of heads.

### 3.1.1. Multi-Head Attention with Projection

- $\overline{W} \in \mathbb{R}^{d \times d/H}$
- $\overline{b},\ \overline{u}^{(i)} \in \mathbb{R}^{d/H}$

$$\overline{v}_t^{(i)} = tanh\left(h_t^{\mathrm{T}}\overline{W} + \overline{b}^{\mathrm{T}}\right)\overline{u}^{(i)}$$

$$\overline{\alpha}_t^{(i)} = \frac{exp\left(\overline{v}_t^{(i)}\right)}{\sum_{t=1}^{T'} exp\left(\overline{v}_t^{(i)}\right)} \tag{3}$$

### 3.1.2. Multi-Head Attention with Split

- $h_t = \left[\mathbf{h}_t^{(1)\mathrm{T}}, \mathbf{h}_t^{(2)\mathrm{T}}, \dots, \mathbf{h}_t^{(H)\mathrm{T}}\right]^{\mathrm{T}},\ \mathbf{h}_t^{(i)} \in \mathbb{R}^{d/H}$
- $\overline{W}^{(i)} \in \mathbb{R}^{d/H \times d/H}$
- $\overline{b}^{(i)},\ \overline{u}^{(i)} \in \mathbb{R}^{d/H}$

$$\overline{v}_t^{(i)} = tanh\left(h_t^{(i)\mathrm{T}}\overline{W}^{(i)} + \overline{b}^{(i)\mathrm{T}}\right)\overline{u}^{(i)}$$

$$\overline{\alpha}_t^{(i)} = \frac{exp\left(\overline{v}_t^{(i)}\right)}{\sum_{t=1}^{T'} exp\left(\overline{v}_t^{(i)}\right)} \tag{4}$$

We apply the weights $\overline{\alpha}_t^{(i)}$ obtained by Equations (3) and (4) to the Equation (1) and define the resulting vectors as $\overline{\alpha}_t^{MP}$ and $\overline{\alpha}_t^{MS}$, which denote the attention weights by Multi-head attention with Projection and Multi-head attention with Split respectively. In the projection method, the output of the LSTM is involved in all the subspaces, but the speaker information may be lost by if the feature dimension is reduced by $d/H$. In the split method, the important parts are not missed because the LSTM output is divided by $d/H$ and then passes through the attention layer. However, the divided part is only used in the generation of different attention weights for each LSTM output of the subspace. To compensate for the disadvantages of the two multi-head attention methods, we herein propose a method to build new measures by combining two attention weights. A new alpha vector was obtained by concatenating two multi-head attention weights as follows:

$$\overline{\alpha}_t^{MC} = \left[ \begin{array}{c} \overline{\alpha}_t^{MP} \\ \overline{\alpha}_t^{MS} \end{array} \right] \tag{5}$$

Here, the alpha matrix $\overline{\alpha}_t^{MC}$ has the dimension 2 *by H*. The new importance vector $\beta_t$ of the new alpha was calculated using the softmax function as shown in Equation (6) and was the *j* index of the concatenated attention weights. Therefore, $\beta_t$ is an additional weight that was used to focus on the more important parts of the multi-head attention.

$$\beta_{t,\,j} = \frac{exp\left(\overline{\alpha}_{t,\,j}^{MC}\right)}{\sum_{j=1}^{2} exp\left(\overline{\alpha}_{t,\,j}^{MC}\right)} \tag{6}$$

$$e^{MC} = \sum_{t=1}^{T'}\left(\sum \overline{\alpha}_t^{MC} \odot \beta_t\right) \odot h_t \tag{7}$$

In the proposed multi-head projection and split combination method, the new attention weight was generated by element-wise multiplication of two vectors $(\overline{\alpha}_t^{MC}, \beta_t)$ and summation for the row

as $\sum \overline{\alpha}_t^{MC} \odot \beta_t$. Finally, the new embedding was obtained by using the attention weights and the LSTM outputs in Equation (7). The process of creating a new embedding with the combination of two multi-head attention is shown in Figure 4.

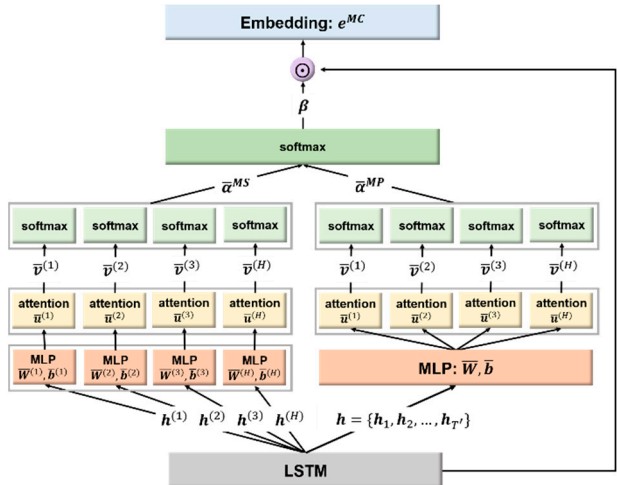

**Figure 4.** Block diagram of the proposed embedding extraction by combination of the projection and split multi-head attentions; the embedding $e^{MC}$ is obtained using the new attention weight $\beta$ which is generated with combination of $\overline{\alpha}^{MS}$ and $\overline{\alpha}^{MP}$. The attention layer output $\overline{v}^{(i)}$ is calculated with the trainable parameters $\overline{W}$, $\overline{b}$ and $\overline{u}$.

### 3.2. Single-Head and Multi-Head Combination

The multi-head attention methods show a more effective performance in speaker recognition systems than the single-head method. However, the essential speaker traits might deteriorate when splitting or projecting the elements in the multi-head methods. Therefore, we propose a method to capture the global dimension with single-head attention and capture the local dimension with multi-head attention. The embeddings $e^{SM}$ obtained from single-head and multi-head attention layers were combined, as shown in Equation (8); subsequently, the proposed new embedding had a dimension of $d \times 2$, where SM represents Single-head and Multi-head combination. We can create multiple attention weights by splitting or projecting the dimensions of the neural network. Therefore, the performance was evaluated by combining the single-head and these two methods and denoted as SM-S (Single/Multi-Split) and SM-P (Single/Multi-Projection) according to the technique used to convert to the subspace. The embeddings $e^{SM-S}$ and $e^{SM-P}$ use $e^{MS}$ and $e^{MP}$ respectively as the embedding of the multi-head attention $e^M$ in Equation (8). The proposed combinations with single-head and multi-head attention are depicted in Figure 5.

$$e^{SM} = \begin{bmatrix} e \\ e^M \end{bmatrix} \tag{8}$$

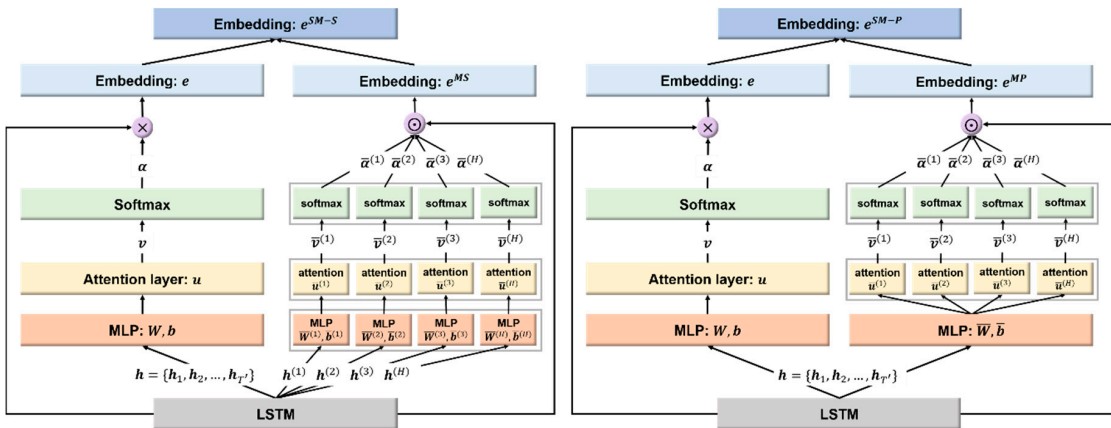

**Figure 5.** Block diagram of the proposed embedding extractions; $e^{SM-S}$ and $e^{SM-P}$ indicate the embeddings by combination of single-head and split-based multi-head attention (**left**), and combination of single-head and projection-based multi-head attention (**right**) respectively. $\overline{\alpha}^{(i)}$, $\overline{v}^{(i)}$, and $h$ denote attention weight, attention layer output, and LSTM output respectively.

## 4. Experiments

### 4.1. Dataset

We used the VoxCeleb1 [22] database, which was constructed by collecting interview video clips of celebrities on YouTube, for speaker recognition and face detection research. The audio data comprise voice signals sampled at 16 KHz uttered for 4 s to 20 s collected 'in the wild' with a range of recording environments as more casual conversation in which laughter, background noise, etc. are observed. The VoxCeleb1 database includes 153,516 utterances from 1251 speakers. The development and test sets were divided for the verification task—148,642 samples from 1211 speakers were used for model training and 4874 samples from 40 speakers were used for inference.

### 4.2. Model Architecture

We trained the UBM and total variability matrix (TVM) with 39-dimensional MFCC. The UBM with 1024 components and a TVM of 400 dimensions were extracted from a 400-dimensional i-vector. In this study, the similarity scores were obtained using Probabilistic Linear Discriminant Analysis (PLDA) using the i-vector and embedding. PLDA is a representative measure for computing similarity between speakers, showing the probability that two utterances are from the same or different speakers based on the likelihood. Dimensional reduction was not applied to the features used in the PLDA training. The proposed model architecture for embedding extraction comprises three parts, which are as follows:

(1) The LSTM-based model receives acoustic features as input and converts them to frame-level features.
(2) The attention layer converts frame-level features to utterance-level features.
(3) GE2E calculates loss during parameter optimization.

#### 4.2.1. LSTM-Based Speaker Verification Model

From a 25-ms frame of input speech with a 12.5-ms moving interval, a 40-dimensional Mel-filter bank feature was extracted. The extracted Mel-filter bank features from 80 neighboring frames were concatenated and placed as the input of LSTM. LSTM comprises 3 layers with 768 nodes, and a 256-dimension projection layer was added. We used the PyTorch [23] to implement the model networks for the experiments.

### 4.2.2. Generalized End-to-End (GE2E) Loss

In the GE2E, the loss function is defined as a similarity matrix designed as a high similarity score between samples uttered by the same speaker and a low similarity score between utterance samples from different speakers. The average of the utterances of each speaker is obtained and used as the speaker's centroid (voiceprint). The loss function is defined by adding the similarity calculated with the centroid of the true speaker and negative form of the similarity calculated with the centroid of the other speaker. This loss function has the effect of placing the speaker embedding close to the centroid of the true speaker and away from the centroid of other speakers. A similarity matrix is generated using M utterances of N speakers per iteration, and in this study, we used N = 8 and M = 8.

### 4.2.3. Attention Layer

The attention layer generates the attention weights by setting the trainable parameters $W$, $b$, and $u$. The final output dimension of LSTM is 256, and the number of heads is set to four. When single-head and multi-head embeddings were concatenated, the dimension of the resulting embedding was doubled, that is, 512. Therefore, the projection layer node was used in 128 dimensions to fit the dimension of the final embedding to 256 dimensions.

### 4.3. Experimental Results

Table 1 presents a comparison of the proposed method with the baseline system in terms of speaker verification performance. We evaluated the performance using the EER measure, which is typically used in speaker verification. The EER is the point where the False Rejection Rate (FRR) and the False Alarm Rate (FAR) are same. The FRR is the percent of incorrectly rejected true users and it is identical to False Negative Rate (FNR). The FAR is the percent of imposter incorrectly matched to a true user, which is also called as False Positive Rate (FPR). By applying single-head attention to the LSTM model, the EER is increased to 5.82% with a 3.37% relative improvement compared to the baseline system. Furthermore, when multi-head attention is applied, the projection and split methods show 5.57% and 5.50% EERs, which are 4.30% and 5.50% relative improvements, respectively, as compared to the single-head model. The EER results of all the systems are presented in Figure 6.

**Table 1.** Equal Error Rate of speaker verification systems.

| Models | | Labels | EER [%] |
|---|---|---|---|
| i-vector/PLDA [4] | | IV | 6.02 |
| LSTM (no attention) [9] | | LSTM | 5.63 |
| Single-head [11] | | SH | 5.82 |
| Multi-head with Projection [15] | | MP | 5.57 |
| Multi-head with Split [16] | | MS | 5.50 |
| Projection & Split Multi-head Combination | (proposed) | MC | 5.53 |
| Single & Multi-head with Split Combination | (proposed) | SM-S | 5.39 |
| **Single & Multi-head with Projection Combination** | (proposed) | **SM-P** | **4.45** |

Among the combination methods proposed herein, the method using the new weight obtained by combining two multi-head attention models has a 5.53% EER, which is slightly worse than that of the systems where only split-based multi-head attention is used. This shows that the multi-head methods do not effectively complement each other to improve recognition performance. The combination method of embeddings generated by the single-head and split-based multi-heads showed a 5.39% EER. Furthermore, the combination of single-head and projection-based multi-head methods showed the best performance with 4.45% EER. Compared to the models in which only the single-head or multi-heads were used, our proposed methods showed significant improvement in terms of the relative EER. Based on these results, we conclude that the inclusion of the global feature of the neural network-based frame-level representation by the addition of the single-head model to the multi-head

model in our combination methods is highly effective. The difference between the weights generated from the single-head and multi-head attentions is illustrated in Figure 7.

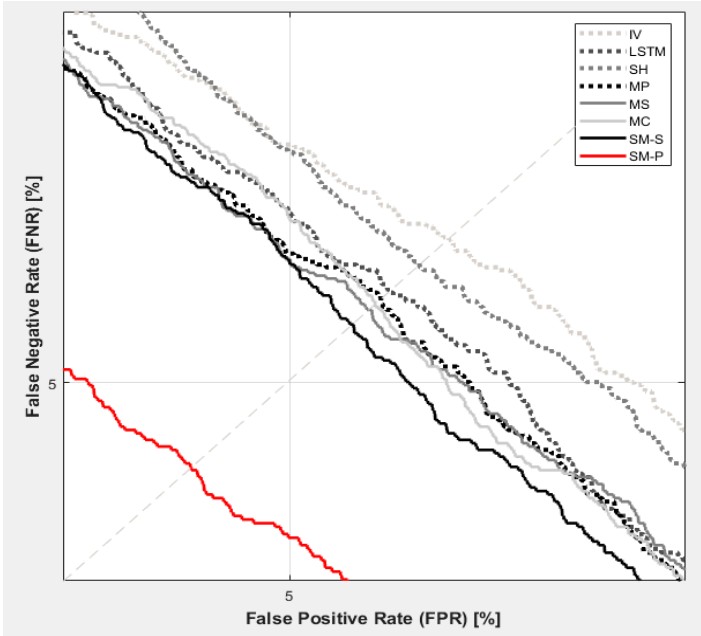

**Figure 6.** Detection Error Tradeoff including the baseline system of i-vector and speaker embeddings and two proposed methods.

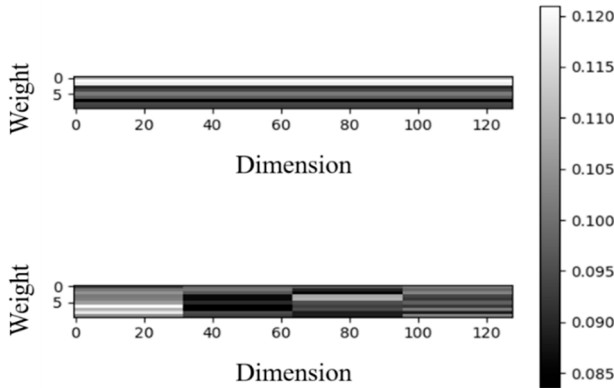

**Figure 7.** Difference in weight of single/multi-head attention models.

The parameters $\alpha_t$ generated by the single-head attention model indicate the same importance for all the elements of the embedding of each frame in up of Figure 7, and the parameter $\alpha_t^M$ in the multi-head attention model shows the different importance for each part divided by the number of heads ($H = 4$) in down of Figure 7. As can be seen, because single-head attention has the same weight in all dimensions, all the elements of the obtained embedding are evenly used for speaker classification. However, in the case of multi-head attention, the divided parts of the embedding are effectively utilized and contribute differently to the speaker classification performance based on the calculated importance weights.

As the multi-head methods lose essential speaker features of the frame-level through the process of projection or splitting, the importance reflecting the entire dimension would be effective to improve speaker verification accuracy by combining the single-head attention method.

## 5. Conclusions

We proposed advanced methods for extracting new embeddings by compensating the disadvantages of the single-head and multi-head attention methods. The combination method of the single-head and split-based multi-head attention showed a 5.39% EER. When the single-head and projection-based multi-head attention methods were combined, the speaker recognition accuracy increased to 4.45% EER, showing the best performance herein. Our experimental results demonstrate that the attention mechanism reflects the speaker's properties more effectively than average or statistics pooling, and the speaker verification system could be further improved by employing combinations of different attention techniques.

**Author Contributions:** Data curation, A.B.; Formal analysis, A.B.; Funding acquisition, W.K.; Investigation, A.B.; Methodology, A.B.; Project administration, W.K.; Writing—original draft, A.B.; Writing—review and editing, W.K. All authors have read and agreed to the published version of the manuscript.

**Funding:** This work was supported by Incheon National University Research Grant in 2017.

**Conflicts of Interest:** The authors declare no conflict of interest.

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
