# Peer review of "Speaker Verification Employing Combinations of Self-Attention Mechanisms"

_electronics, doi:10.3390/electronics9122201_

Round 1

Reviewer 1 Report

The paper presents a method for speech recognition, in which the construction of the embedding vector is obtained by combining a single-head and a multi-head method. From the case study, the authors note that the multi-head method by itself does not lead to significant improvements over the single-head, which indicates that some information is lost in vectorization. Conversely, by combining the two methods it is possible to obtain a significant improvement in performance.

Remarks

The authors do not provide the rationale of the method with arguments of a general nature, so the results obtained on the benchmark, although this is widely used in literature, leave doubts about what the results on other datasets may be.   Minor Points It would be worth briefly illustrating the tools used in the method. In particular, LSTM networks, MFCC, GE2E, UBM, PLDA deserve to be briefly illustrated. Finally, the formula for calculating the EER should be provided.

Reviewer 2 Report

This paper presents an effective speaker verification framework that's demonstrated to out perform several recent methods, but I do have several comments,

1) I think adding a little bit figure for visual explanation of single head and multi-head attention would be very helpful, currently it's a little hard to follow.

2) The fonts of equations are too big, please adjust them according to the text. Also when abbreviation appears in the first place, please add the full name, e.g. the ERR in abstract.

3) I'm not sure if those equations in the related work are fully related to the proposed methods (seems like they are), usually we prefer not to see much equations in the related work part. So basically please focus more on the development of past work on your topics, such as telling a story of the pros and cons of recent studies, and how they are related to your method, and don't put too many equations here. If those equations are important, please move them to the section 3, where you would have a more organized explanation of your formulations.

More importantly, explanation of equations are poor, please add as much details as possible about notations, and what they mean, etc, since currently it is extremely hard to me to follow since a lot of notations has not been explained well.

4) Fig.3 and 4, it is hard to tell the difference and link between them , please try to highlight so people can easily appreciate the benefits of your methods. And, captions could be as much detailed as possible, explaining what's going on in the figure.

5) Table 1. are those methods being compared from the reference, if yes please add citation mark.

Round 2

Reviewer 2 Report

The overall presentation of the paper has been improved, but there are still a few concerns. For example, the figures have been compressed, so please use vector format so they are clearer, especially fig. 3,4,5.

Moreover, captions are still not enough, please add more highlight, e.g. in figure 5, explain all notations not only in the caption, but also in the text.
